# Tracking calcium dynamics from individual neurons in behaving animals

Thibault Lagache[1,2¤]*, Alison Hanson[1,2,3], Jesús E. Pérez-Ortega[1], Adrienne Fairhall[4,5], Rafael Yuste[1,2,6]

**1** Department of Biological Sciences, Columbia University, New York, New York, United States of America, **2** Marine Biological Laboratory, Woods Hole, Massachusetts, United States of America, **3** Department of Psychiatry, New York State Psychiatric Institute, Columbia University, New York, New York, United States of America, **4** Department of Physiology and Biophysics, University of Washington, Seattle, Washington, United States of America, **5** UW Computational Neuroscience Center, University of Washington, Seattle, Washington, United States of America, **6** Donostia International Physics Center, San Sebastian, Spain

¤ Current address: Department of Cell Biology and Infection, Institut Pasteur, Paris, France
* thibault.lagache@pasteur.fr

**Data Availability Statement:** Code is freely available from the open-source platform Icy (icy.bioimageanalysis.org): http://icy.bioimageanalysis.org/plugin/elastic-motion-correction-concatenation-emc2-of-tracks/ Movies used in this

## Abstract

Measuring the activity of neuronal populations with calcium imaging can capture emergent functional properties of neuronal circuits with single cell resolution. However, the motion of freely behaving animals, together with the intermittent detectability of calcium sensors, can hinder automatic monitoring of neuronal activity and their subsequent functional characterization. We report the development and open-source implementation of a multi-step cellular tracking algorithm (Elastic Motion Correction and Concatenation or EMC$^2$) that compensates for the intermittent disappearance of moving neurons by integrating local deformation information from detectable neurons. We demonstrate the accuracy and versatility of our algorithm using calcium imaging data from two-photon volumetric microscopy in visual cortex of awake mice, and from confocal microscopy in behaving *Hydra*, which experiences major body deformation during its contractions. We quantify the performance of our algorithm using ground truth manual tracking of neurons, along with synthetic time-lapse sequences, covering a wide range of particle motions and detectability parameters. As a demonstration of the utility of the algorithm, we monitor for several days calcium activity of the same neurons in layer 2/3 of mouse visual cortex *in vivo*, finding significant turnover within the active neurons across days, with only few neurons that remained active across days. Also, combining automatic tracking of single neuron activity with statistical clustering, we characterize and map neuronal ensembles in behaving *Hydra*, finding three major non-overlapping ensembles of neurons (CB, RP1 and RP2) whose activity correlates with contractions and elongations. Our results show that the EMC$^2$ algorithm can be used as a robust and versatile platform for neuronal tracking in behaving animals.

study (avi and tiff files) can be downloaded from the BioStudies platform (https://www.ebi.ac.uk/biostudies/studies/S-BSST428).

**Funding:** R.Y. was supported by the NSF (CRCNS 1822550), the NEI (R01EY011787), the NIMH (R01MH115900), and Vannevar Bush Faculty Award (ONR N000142012828). T.L. was supported by the Fondation pour la Recherche Médicale (https://www.frm.org/) and the Philippe Foundation (https://www.philippefoundation.org/). A.H. was supported by the NIMH (T32MH018870). J.P.-O. was supported by the CONACYT (CVU365863). ALF was supported by NSF (CRCNS 1822550), the Simons Foundation Collaboration for the Global Brain (542975SPI) and the Weill NeuroHub (https://www.weillneurohub.org/). The funders had no role in study design, data collection and analysis, decision to publish, or preparation of the manuscript.

**Competing interests:** The authors have declared that no competing interests exist.

## Author summary

Calcium imaging of neuron populations has enabled mapping the neuronal circuits that control animal behavior. However, animal movement, together with the intermittent detectability of calcium sensors, hinders the automatic tracking of individual neuron activity. Here we introduce a novel algorithm and open-access software to track the position of individual neurons in a calcium imaging movie in behaving animals. To handle the motion and the deformation of the animal our method combines state-of-the art algorithms to track neurons, with algorithms to estimate the deformation and predict the positions of neurons when they are silent and undetectable. Our method and software are robust and versatile in various animal models, from two-photon imaging of mouse visual cortex over days, to the highly deforming *Hydra*. Efficient image analysis and software for monitoring the activity of neuron populations in a wide range of animal models are needed to fully reconstruct the activity of neural circuits and study the emergent functional properties of neuronal ensembles that control animal state and behavior.

This is a *PLOS Computational Biology* Methods paper.

## Introduction

Measuring the activity of neuronal populations in freely behaving animals can help a detailed understanding of how neural circuits integrate external information, compute, learn and control animal behavior. Calcium imaging has become widespread for measuring single neuron activity as it is non-invasive and allows the simultaneous measurement of hundreds to thousands of cells, with single cell resolution [1]. Moreover, monitoring single neuron activity in freely moving animals such as rodents can be achieved with miniaturized microscopes attached to the head [2]. However, technical limitations of current microscopy techniques and of mathematical analysis hinder a more complete imaging and analysis of entire brains. Alternative strategies consist of monitoring single neuron activity in targeted brain regions of rodents using two-photon microscopy [3], or imaging the nervous system of a smaller animal, one that can fit entirely within a microscope's field of view, such as *Caenorhabditis elegans* [4], *Hydra* [5], Zebrafish [6] or *Drosophila* larvae [7]. An advantage of simple model organisms is that they contain many fewer neurons than mammals and have a limited repertoire of behaviors [8] that may be entirely characterized in the near future.

Aside from the difficulties in imaging entire nervous systems with high temporal and spatial resolution [3, 6, 9], an important bottleneck in analyzing calcium imaging data is to achieve robust and automatic tracking of individual cells' position while the animal is behaving. Single-cell tracking is challenging for three main reasons: First, there may be a large number of cells in a cluttered environment. Therefore, false positives and negatives during single cell detection and localization impede the association of detected neurons between successive time frames and call for more elaborate tracking algorithms. Second, neurons can remain undetectable over large periods of time because calcium sensors may be significantly brighter than background only when neurons are firing. Third, tracking methods in behaving animals have to handle animal motion and body deformations [10]. To tackle these issues, numerous hardware solutions have been proposed, including fixation of animal (usually the head) [3, 6, 11], high-speed motorization of microscopes to track animal movements [4, 12] and dual-color

labeling of neurons with a calcium-insensitive probe that can be detected and localized even when neurons are not firing [4, 9]. However, even when (some of) these solutions are implemented, the residual motion of neurons, the limited spatial resolution of the microscope and the intermittency of the fluorescence signal (when single-channel calcium imaging is used) hinder the robust tracking of single neuron activity, particularly over long periods of time. Therefore, elaborate post-processing of acquired movies is required. Most current image processing methods consist of registering images (volumes) to reference image(s) (volume(s)) using either the local fluorescence intensity of images [13, 14], or the extracted neuron positions directly [4, 10, 15, 16]. Then, additional image processing, such as non-negative matrix factorization [17, 18], might be required for demixing and denoising cellular calcium activity.

Despite these software developments, the difficult implementation of hardware solutions, such as the dual-labeling and imaging of calcium-insensitive probes, together with the significant deformability of challenging animal models such as *Hydra*, prevent the robust automatic tracking of single neurons in many experiments. Tracking has then to be performed manually [5] or semi-manually [19]. This limits the analysis to a few hundred frames, introduces operator bias and, ultimately, hinders our understanding of the functional organization of nervous systems.

To robustly track particles with intermittent detectability (neurons) in a cluttered and deforming environment, we report the development of an algorithm named Elastic Motion Correction and Concatenation (EMC$^2$). EMC$^2$ is based on the versatile framework of single-particle-tracking (SPT), enabling the robust monitoring of single neuron activity in most, if not all, animal models. In contrast to traditional SPT, EMC$^2$ does not set expected priors for particle motion (typically diffusion and/or linear motion). Instead, it uses information about local motion and deformation from detectable and tracked particles in the neighborhood of undetectable particles. EMC$^2$ is therefore more versatile, and does not require motion priors or heuristics to close tracking gaps. In addition, for the local tracking of detectable particles, EMC$^2$ uses a probabilistic method and is therefore robust to very cluttered conditions. We validate the robustness and accuracy of EMC$^2$ with manual tracking of neurons in two calcium imaging datasets from behaving animals. Our first dataset consists of two-photon calcium imaging of few tens of neurons in the visual cortex of awake mice. We show that our algorithm accurately tracks the limited motion of single neurons in the two-photon field-of-view, enabling the fast analysis of long recordings of individual neuron activity. We then monitor single neurons of *Hydra's* nervous system while the animal is behaving and deforming. *Hydra* imaging datasets represent perhaps the worst possible scenario for tracking purposes, since animals can have major changes in body size with non-isometric deformations. We also quantify the performance of EMC$^2$ using simulations of fluorescence time-lapse sequences with different types of motion (confined diffusion, linear displacement and elastic deformation), and show that EMC$^2$ outperforms state-of-the art tracking algorithms.

After integrating EMC$^2$ in an open-source and freely available platform Icy [20] (icy. bioimageanalysis.org), we explore the utility of the algorithm in two experimental scenarios. We first monitor the activity of single neurons in two-photon calcium imaging of layer 2/3 of mouse visual cortex over multiple days and show that there is an important turnover of active neurons and that very few neurons remain active across days. We then track complete neuron activity in behaving *Hydra*, and find functional clustering of individual neurons into co-active ensembles [21]. Consistent with previous observations [5], we find that *Hydra* contains three main neuronal ensembles (CB, RP1 and RP2); and, after mapping the positions of individual neurons from each ensemble, we also confirm that these ensembles are not overlapping, i.e., they do not share neurons, and that they are correlated with contraction bursts and elongation behaviors.

These results demonstrate that EMC$^2$ is an effective and versatile tracking algorithm for the tracking of single neuron activity in calcium imaging of living animals. Robust tracking constitutes a prerequisite for the statistical analysis of the functional organization of neural circuits, the description of emergent computational units such as neuronal ensembles, and ultimately, for the understanding and prediction of animals' adaptive behavior.

## Results

### 1 Statistical mapping of neuron positions versus single-particle-tracking

Most of the methods that have been developed for tracking single neuron activity in the well-documented animal model *C. Elegans* are based on the statistical mapping of neuron positions to a reference set of coordinates within the animal [10, 16, 22]. An important advantage of these methods is their robustness to the length of the analyzed time-lapse sequence, as the different images are registered independently from each other to the reference set of positions. In these methods, reference positions are extracted from reference frames and cell identities can be obtained either from stereotyped fluorescent color maps of all neurons (*NeuroPAL* [23]) or online cell atlases (e.g. *WormAtlas* [24] and *OpenWorm* [25]). Therefore, mapping methods heavily rely on the stable repertoire of neurons (position and/or type) within single worms, when mapping is used for tracking neurons along time [10, 18, 26], and even across different worms when different animals or strains are compared [16, 22].

Unfortunately, in *Hydra*, the number and position of neurons differ from one animal to another [27]. Moreover, even within a single animal, the accurate mapping of a reference set of neuron positions is prevented by the important and continual deformations of the animal (typically, the length of the animal is reduced by more than half during longitudinal contraction), and the 2D imaging of transparent 3D tissues that induces apparent changes in neuron positions even between stereotypical poses of the animal. Finally, while the aforementioned mapping methods used in *C. Elegans* can accommodate moderate changes in the total number of neurons between frames due to missing or spurious detections (counting noise), the intermittent and sparse detectability of neurons in calcium imaging, without a reference fluorophore like calcium-insensitive red fluorescent proteins (RFP) [10], definitely hinders the applicability of mapping methods.

Therefore, to track neurons in *Hydra*, we used the SPT framework. SPT methods sequentially link cell detections through time [8], with no need for a reference set of positions. They are more versatile than statistical mapping, and can handle the large deformations of *Hydra* and the intermittent detectability of cells in calcium imaging.

### 2 Limitations of standard SPT Algorithms

Most SPT algorithms rely on the automatic detection of particles (cells, molecules. . .) that are significantly brighter than the noisy background in each frame of the time-lapse sequence (see [28] and [29] for review) and, subsequently, the linking of detections between frames corresponding to the reconstruction of coherent particle trajectories (see Table 1). The prevalence of false positives (i.e. background signal) and negatives (i.e. missing detections) in the detection of particles, together with the influence of high particle density and stochastic dynamics have, over the last two decades, motivated the development of algorithms that go beyond naïve tracking methods that simply associate nearest-neighbor detections between consecutive time frames (see Table 1 and [30, 31] for a review of existing methods). Indeed, the erroneous association of one detection to a track, or the premature ending of a track due to missing detection (s), can lead to important error propagation as detections are sequentially associated to existing tracks.

**Table 1. Tracking methods in bio-imaging.**

| Algorithm | Type | Detection | Linking | Gap closing | Pros | Cons | Freely available | Ref. |
|---|---|---|---|---|---|---|---|---|
| Sage et al. | Global | None | Energy minimization | Yes | Global. | Designed for one or few sparse particles | ImageJ plugin | [53] |
| Bonneau et al. | Global | None | Energy minimization | Yes | Global. Robust gap closing with minimal-path algorithm | Designed for few sparse particles. High computational load. | No | [54] |
| NeRVE | Detect & Mapping | Watershed Segmentation | Point-set registration & clustering–Animal deformation estimated with elastic transformations | Yes | Robust to dense packing of particles. Handles non-linear deformations. Time-independent (i.e. robust even in long time-lapse sequences) | High computational load. Not robust to many missing detections and long gaps in highly deforming environments. | Matlab GUI | [10] |
| fDLC | Detect & Mapping | Watershed Segmentation | Point-set registration to reference set of positions–learning of animal deformation | Yes | Robust to dense packing of particles. Handles non-linear deformations. Time-independent | Not robust to many missing detections and long gaps in highly deforming environments. | Python (Github repository) | [16] |
| CRF_ID | Detect & Mapping | Gaussian mixture model fitting | Point-set registration to reference & temporally-nearby frames–Use of graphical model (neighbors) to predict identities | Yes | Robust to dense packing of particles. Handles non-linear deformations. Time-independent | High computational load. Not robust to many missing detections and long gaps in highly deforming environments. | Matlab (Github repository) | [22] |
| Mosaic | Detect & Link | Gaussian Convolution & Thresholding | Global distance minimization | Yes | Fast. Accounts for spot intensity and size in distance computation. | Gap closing does not handle large motion. Not robust in very cluttered conditions. | *Particle Tracker* plugin (ImageJ) | [32] |
| TrackMate | Detect & Link | Wavelet transformation or Gaussian convolution & thresholding | Global distance minimization | Yes | Fast. Handles split & merge events. | Gap closing does not handle large, non-linear motion. Many user-defined parameters | *TrackMate* plugin (ImageJ) | [33] |
| eMHT | Detect & Link | Wavelet transformation & thresholding | Probabilistic (Multiple Hypothesis) | Yes | Robust in cluttered environment. Few user-defined parameters | Slower than global distance minimization. Cannot close large gaps (> ~5 frames) due to computational load | *Spot Tracking* plugin (Icy) | [30] |
| MAP-4D-DAE | Detect & Link | Not specified | Probabilistic (Multiple Hypothesis) + Autoencoding for particle motion modeling | Yes | Robust in cluttered environment. Few user-defined parameters. Handles non-linear deformations | Slower than global distance minimization. Cannot close large gaps (> ~5 frames) due to computational load | No | [10] |

One category of elaborated tracking algorithms is based on global distance minimization (GDM) between all pairs of detections in consecutive time frames. The distance measure between detections can be simply the Euclidean distance, or can additionally take into account the similarity of the intensity and/or shape of the detected particle [32]. To handle possible missing and false detections, heuristics for track termination and initiation are defined by the user [33]. GDM methods are fast and robust, but user-defined parameters that regulate the ending, the initiation and the fusion of tracks hinder their applicability in very cluttered conditions with numerous missing and spurious detections [30]. Moreover, the limitations of current particle motion models (confined diffusion and linear directed motion [32–34]) prevent the robust estimation of particle positions when they remain undetectable over long periods of time, as for sparsely firing neurons in calcium imaging. This limits the ability to close gaps in the trajectories of tracked particles to a very few frames.

Probabilistic methods are an attractive alternative to GDM methods, even if their computational cost is higher. Probabilistic algorithms model both the stochastic motion of the particles and their detectability, and then compute the optimal tracking solution by maximizing the model likelihood of observed detections [30, 35, 36]. The gold standard of probabilistic association is multiple hypothesis tracking (MHT) in which one computes all the possible tracking solutions over the entire time-lapse sequence, before inferring the tracking solution that maximizes the observation's likelihood. However, MHT is generally not computationally tractable. Approximate solutions that iteratively compute a nearly optimal solution over a limited number of frames (typically up to 5) have been proposed [30]. As probabilistic methods model the particles' detectability, they are usually more robust than GDM methods in cluttered conditions. However, the small number of frames considered when approximating the MHT solution, together with limited particle motion models (again diffusion and/or linear displacements [30, 35]), reduce the capability of probabilistic algorithms to close large tracking gaps and keep track of particles' putative position when they remain undetectable over many frames.

## 3 EMC$^2$ Algorithm

To increase the capability of SPT algorithms to track single particles with intermittent detectability in cluttered and deforming environments, we have developed EMC$^2$. This multi-step algorithm and software is particularly well-suited for tracking single neuron activity with calcium imaging in a behaving animal. The EMC$^2$ algorithm can be decomposed into four main steps (Fig 1).

First, bright spots (e.g. firing neurons in calcium imaging sequence) are automatically detected with a robust method based on wavelet decomposition of time-lapse sequence and statistical thresholding of wavelet coefficients (Materials and Methods). Second, detected spots are linked into single particle trajectories with a state-of-the-art probabilistic algorithm, a variant of multiple hypothesis tracking (eMHT [30]), which is particularly robust in cluttered conditions. Obtained track(let)s correspond to trajectories of detectable particles. However, in many time-lapse sequences such as calcium imaging of neuron activity, tracks would be terminated prematurely when particles switch to an undetectable state (e.g. non-firing neuron) and new tracks would be generated when particles can be detected again (e.g. firing neurons). This creates time-gaps in individual tracks that need to be closed to allow the accurate tracking of each particle's identity over the time-lapse sequence. Thus, the two last steps of our method aim to close gaps in trajectories using information about the motion and deformation of the field of view along the time-lapse sequence. We considered that tracked particles are embedded in a deformable medium (e.g. neurons within tissue) and that local estimation of the deformation of the field-of-view should allow the inference of particles' positions even when they are undetectable.

The third step of EMC$^2$ is therefore the computation of the elastic deformation of the field-of-view at each time using the information contained in tracklets of detectable particles. For this, we used the positions of tracked particles between consecutive time frames as fiducial *source* and *target* points. We then computed the forward and backward elastic deformations of the whole field by interpolating the deformation at any position between fiducials with a thin-plate-spline function. The thin-plate spline is a popular poly-harmonic spline whose robustness in image alignment and point-set matching has been demonstrated [37], and which has been recently applied for automatic neuron registration in time-lapse sequences [10]. In our hands, the Neuron Registration Vector Encoding (NeRVE) method developed to map single neurons in *C. elegans* [10] is unfortunately not sufficiently robust for tracking neurons in

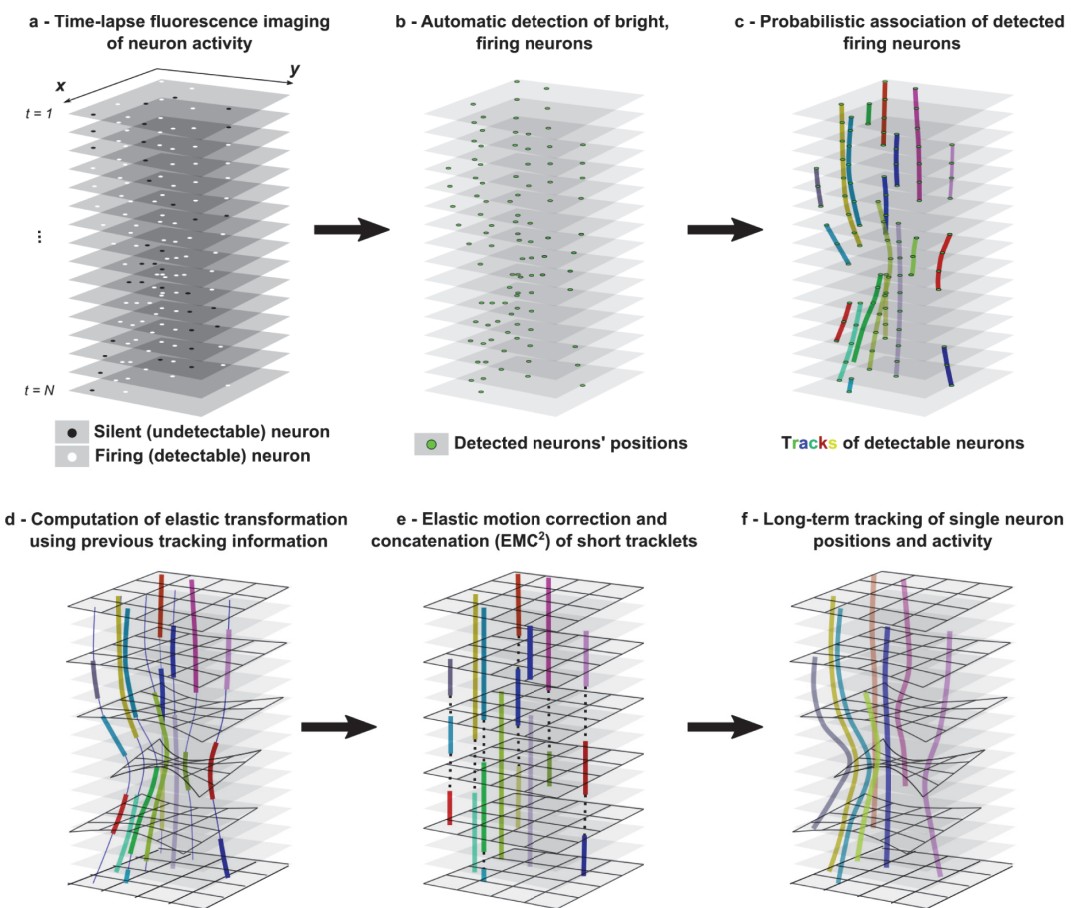

**Fig 1. Multi-step EMC$^2$ for tracking neuron activity in calcium imaging data. a-** Time-lapse imaging (N frames) of intermittent fluorescence activity of a neuron in a deforming environment (e.g. behaving animal). **b-** Fluorescent spots (neurons), that are significantly brighter than background, are automatically detected with a wavelet-based algorithm. **c-** Tracklets of detectable neurons are robustly reconstructed using probabilistic tracking algorithm (eMHT). **d-** Short tracklets of detectable particles are used to compute the elastic deformation of the field of view at each time frame. Associated detections in neuron tracklets are used as fiducials, and the whole deformation is interpolated using a poly-harmonic thin-plate spline function. Forward- and backward-propagated positions of tracklet particle positions are shown with a thin blue line. **e-** After having corrected for the deformation of the field-of-view where neurons are embedded, gaps between the end- and starting-points of tracklets are closed by minimizing the global Euclidean distance between points (dotted line). **f-** Finally, complete single neuron tracks over the time-lapse sequence are obtained by applying the elastic transformation of the field-of-view to concatenated tracklets.

calcium imaging of behaving *Hydra*, as shown in [15]. In addition to the inherent limitations of mapping methods when applied to calcium imaging of *Hydra* (see paragraph "*Statistical Mapping of Neuron Positions versus Single-Particle-Tracking*"), the poor tracking accuracy of the NeRVE method when applied to *Hydra* is also due to the fact that, contrary to *C. elegans*, one cannot map the *Hydra* neuron positions in fixed cylindrical coordinates along the principal axis of the animal. This reflects a much lower level of effective deformation in *C. elegans* and consequently a lower complexity of neuron registration. To increase the robustness and accuracy of single neuron tracking, we rather implement, in EMC$^2$, a local concatenation of short tracklets after having propagated forward and backward the estimated deformation of the field of view.

Therefore, the fourth and last step of our method is the iterative estimation and correction of the elastic deformation of the field-of-view, followed by the optimal concatenation of

tracklets. In this last step, we used the elastic transformation computed with thin-plate splines to propagate forward (and backward) the putative positions of undetectable particles, following the termination of their detectable tracklets (or preceding the initiation of novel tracklets). After having corrected for the elastic deformation of the field-of-view, we then linked short tracklets by minimizing the global distance between the end-points of prematurely terminated tracklets with the starting-points of newly appearing tracklets (Materials and Methods). Finally, single-particle tracks over the time-lapse sequence are obtained by applying the computed elastic transformation to the concatenated tracklets.

Contrary to gap-closing GDM approaches [33], $EMC^2$ contains only two free parameters: the maximal distance between forward-propagated ending-points and backward-propagated starting-points of short tracklets for concatenation, and, to avoid important error propagation, a maximum time-lag between ending- and starting-point candidates. We highlight that concatenated tracklets do not necessarily span the entire time lapse sequence: each track begins with the first detection of its first concatenated tracklet and ends with its last detection of the last concatenated tracklet. Moreover, $EMC^2$ algorithm handles complex natural motions and deformations, contrary to GDM methods that only account for confined or linear motion. $EMC^2$ is therefore more robust and versatile. For the sake of reproducibility and dissemination of our method, we implemented the $EMC^2$ multi-step procedure in the bio-image analysis software suite Icy [20] (http://icy.bioimageanalysis.org/). Icy is an open-source platform that is particularly well-suited for multi-step analysis thanks to graphical programming (plugin *protocols*) where each step of the analysis can be implemented as a *block* with inputs and outputs that can be linked to the other blocks (Fig 2 and Materials and Methods). Our method builds on well-established Icy preprocessing functions for spot detection and tracking.

## 4 Validation of $EMC^2$

**a Manual tracking of calcium dataset in behaving animals.**   To validate the capabilities of $EMC^2$, we first compared the results of our algorithm with manual tracking in calcium imaging sequences of neuron activity in behaving *Hydra* [5]. We used the first 250 frames (25 seconds at 10 Hz) of a time-lapse sequence previously acquired in a genetically-engineered animal [5] and automatically detected the active neurons (bright spots) using the multi-step detection process described in the Materials and Methods. We tracked the detected particles with the eMHT algorithm ([30], implemented in *Spot tracking* plugin in Icy) and obtained short Bayesian tracklets (n = 784 tracklets) for the detected neurons (step 4 of the Icy protocol in Fig 2). We then manually concatenated all the corresponding tracklets, i.e. we closed gaps, and obtained complete neuron tracks (n = 444 tracks). We observed that, before gap closing, tracklets were significantly shorter than concatenated tracks, meaning that many tracklets are indeed terminated prematurely by the undetectability of silent (non-firing) neurons. We measured the accuracy of $EMC^2$ by comparing the computed tracks with those obtained after manual association of tracklets, which we took as an approximate ground truth (see Materials and Methods and S2 Fig). We also measured how tracks obtained with *TrackMate* [33] implemented in Fiji [34] matched this manual ground truth. *TrackMate* is a GDM method, based on optimal linear assignment between closest detections. To handle the gaps in tracking when particles are undetectable, *TrackMate* again uses a GDM algorithm to compute the optimal linear assignment between ending- and starting-points of previously computed tracks. As a result, *TrackMate* uses the same type of gap closing algorithm as $EMC^2$ but without correcting for potential elastic deformation of the field-of-view. Finally, to evaluate how well the algorithm performs to generate the initial set of short tracklets, i.e. to compare the probabilistic eMHT used in $EMC^2$ with the GDM algorithm used in *TrackMate*, we also measured the accuracy of $EMC^2$, but without

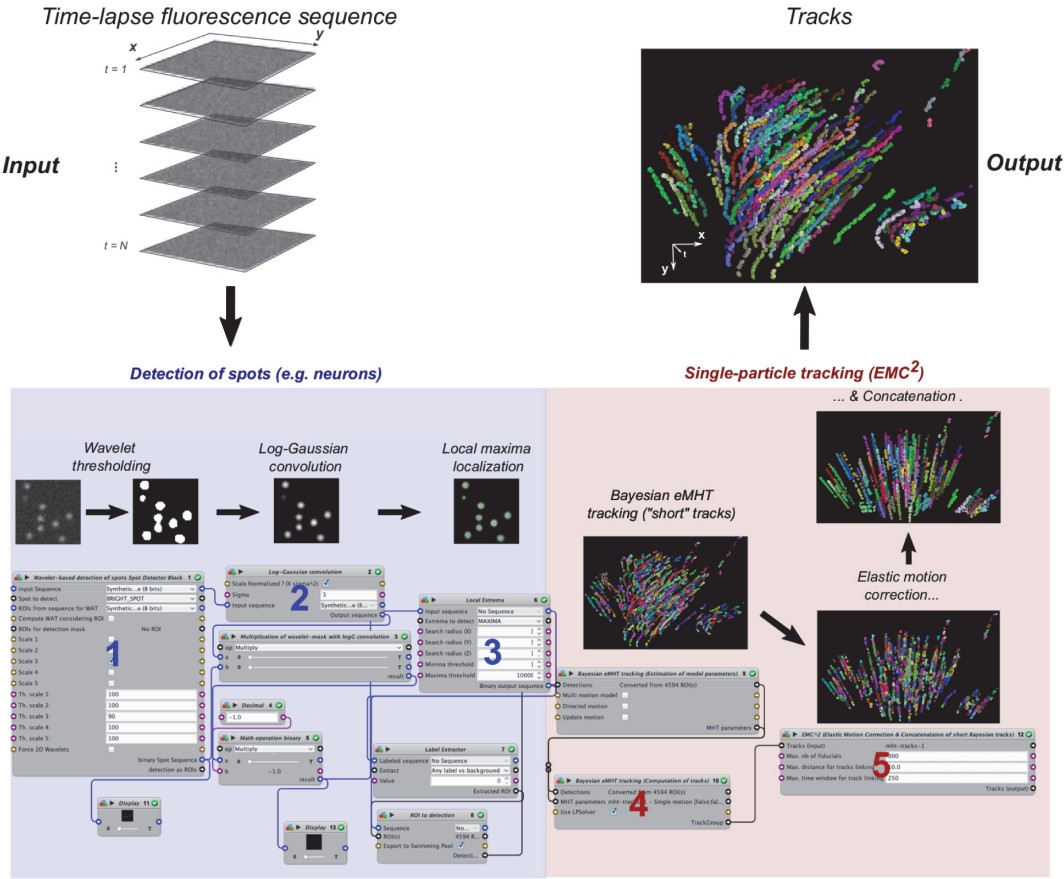

**Fig 2. Implementation of the EMC² algorithm in Icy platform.** Time-lapse sequence of fluorescent particles is the input to a multi-step, automatic protocol in Icy. A first series of blocks, highlighted in blue, detects the position of fluorescent neurons (spots) in each frame of the time-lapse sequence. Block 1 uses the wavelet transform of each image and statistical thresholding of wavelet coefficients to determine spots that are significantly brighter than background. To separate close spots that form clusters in the wavelet-based mask of the image, the thresholded sequence is convolved with a log-gaussian kernel to enhance single spots (block 2), and local maxima algorithm is applied (block 3). A second series of blocks, highlighted in red, computes single particle tracks from computed spot positions. First, the Bayesian tracking algorithm (eMHT) computes tracklets of detectable particles (block 4). Due to fluctuating particle detectability, many Bayesian tracklets are terminated prematurely and new tracklets are created when particles can be detected again. To close detection gaps in single particle tracks, block 5 applies the EMC² algorithm. Final output of the Icy protocol is the collection of single particle tracks over the time-lapse sequence. Tracking protocol can be found here: http://icy.bioimageanalysis.org/protocol/detection-with-cluster-un-mixing-and-tracking-of-neurons-with-emc2/ and is also directly accessible through the search bar of the Icy software (see step-by-step Supplementary Icy tutorial).

elastic motion correction before gap closing. Compared algorithms are summarized in Table 2. First, we found that EMC² (n = 453 tracks, with 410 (90.5%) matched tracks) outperformed TrackMate (n = 474 tracks, with 259 (54.6%) matched tracks) and EMC² without elastic

**Table 2. Tested tracking algorithms in manual validation.**

| Name | Local association of detected particles | Elastic Motion Correction? | Gap closing | % match |
|---|---|---|---|---|
| Manual | Bayesian (eMHT) | No | Manual | 100% (*ground truth*) |
| EMC² | Bayesian (eMHT) | Yes | GDM | 90.5% |
| TrackMate | Global distance minimization (GDM) | No | GDM | 54.6% |
| EMC² without Correction | Bayesian (eMHT) | No | GDM | 54.3% |

correction (n = 514 tracks, with 279 (54.3%) matched tracks). The similar capabilities of Track-Mate and EMC$^2$ without elastic motion correction indicate that Bayesian eMHT and the GDM tracking method perform similarly for local association of detectable spots, but fail at closing longer tracking gaps in deformable media. This highlights the importance of elastic motion correction before the optimal concatenation of short tracks.

Using the same methodology, we compared manual and automatic (EMC$^2$) tracking of single neurons in the less challenging case of two-photon imaging of the visual cortex (layer 2/3) of mice (we used the first day calcium recording from the first animal [38]). Here, the deformation of the field-of-view is much more limited and the motion of embedded neurons can be assimilated to confined diffusion. As expected, we obtained an EMC$^2$ accuracy that was close to 100% (64 correct tracks over 65, i.e. 98.5% accuracy) for a time lapse sequence of 3,700 frames (5 minutes).

**b Synthetic time-lapse sequences.** Manual gap closing in time-lapse sequences is tedious and prone to operator bias. Moreover, the *ground truth*, *i.e.* the identity of each individual neuron along the whole time-lapse sequence, is unknown. Therefore, we designed a reproducible, synthetic approach where we simulated individual neurons' activity and animal motion with different sets of parameters.

We modeled three different types of motion and/or deformation (Materials & Methods): *Confined diffusion*, where blinking neurons diffuse within a confined area (as in two-photon imaging of targeted brain areas), *Linear motion*, where neurons all move together in the same direction at constant velocity, and finally, using deformation fields measured in Hydra experimental data, we simulated naturalistic *Hydra* deformations. We further modeled the intermittent activity of neuronal ensembles with a probabilistic Poisson model. We also modeled the fluorescence dynamics of individual spikes using a parametric curve that we fitted to experimental data. Finally, using neuron positions, firing activity and fluorescence dynamics, we generated synthetic time-lapse sequences using a mixed Poisson-Gaussian noise model ([30] and Materials & Methods).

For confined diffusion (150 simulated tracks, n = 10 simulations), both EMC$^2$ and Track-Mate gave excellent results, with track matches of 93.5% ± 1.6% (144.4 ± 1.8 correct tracks over 154.6 ± 0.9 reconstructed tracks) for EMC$^2$ and 92.9% ± 0.8% for TrackMate (140.3 ± 1.0 correct tracks over 151.0 ± 0.3 reconstructed tracks) (Fig 3A). The good performance of Track-Mate was expected as this algorithm was initially designed to track confined endocytic spots at the cell membrane [33]. In addition to confined motion, TrackMate can also model linear motion of particles when computing the optimal gap closing between short tracks. Therefore, in linear motion simulations (337.6 ± 1.0 simulated tracks, n = 10 simulations), we used Track-Mate with linear motion correction instead of standard confined motion correction. However, even with linear correction, the performance of Trackmate (76.3% ± 0.6% (262.6 ± 1.4 correct tracks over 358.2.0 ± 1.8 reconstructed tracks)) was significantly worse than EMC$^2$'s performance (97.7 ± 0.5% (326.1 ± 1.6 correct tracks over 337.3 ± 1.1 reconstructed tracks)). This difference is due to the different estimation methods that are used in the two tracking algorithms to estimate the direction of tracks: in TrackMate, the estimation of track directions is local, based on the last detection within each short track, whereas the estimation of track direction in EMC$^2$ uses global information provided by neighbouring short tracks and is therefore more robust. Finally, in the third case, we used the deformation field that we estimated over 250 frames within a time-lapse experimental sequence of Hydra (500 simulated tracks, n = 10 simulations, see section 2.a and Materials & Methods). We found that Trackmate had similar performances with (matching score 69.4 ± 1.1%, or 414.8 ± 3.7 correct tracks over 598.2 ± 4.8 reconstructed tracks) or without (72.4 ± 0.9%, or 427.2 ± 3.0 correct tracks over 590.7 ± 3.5

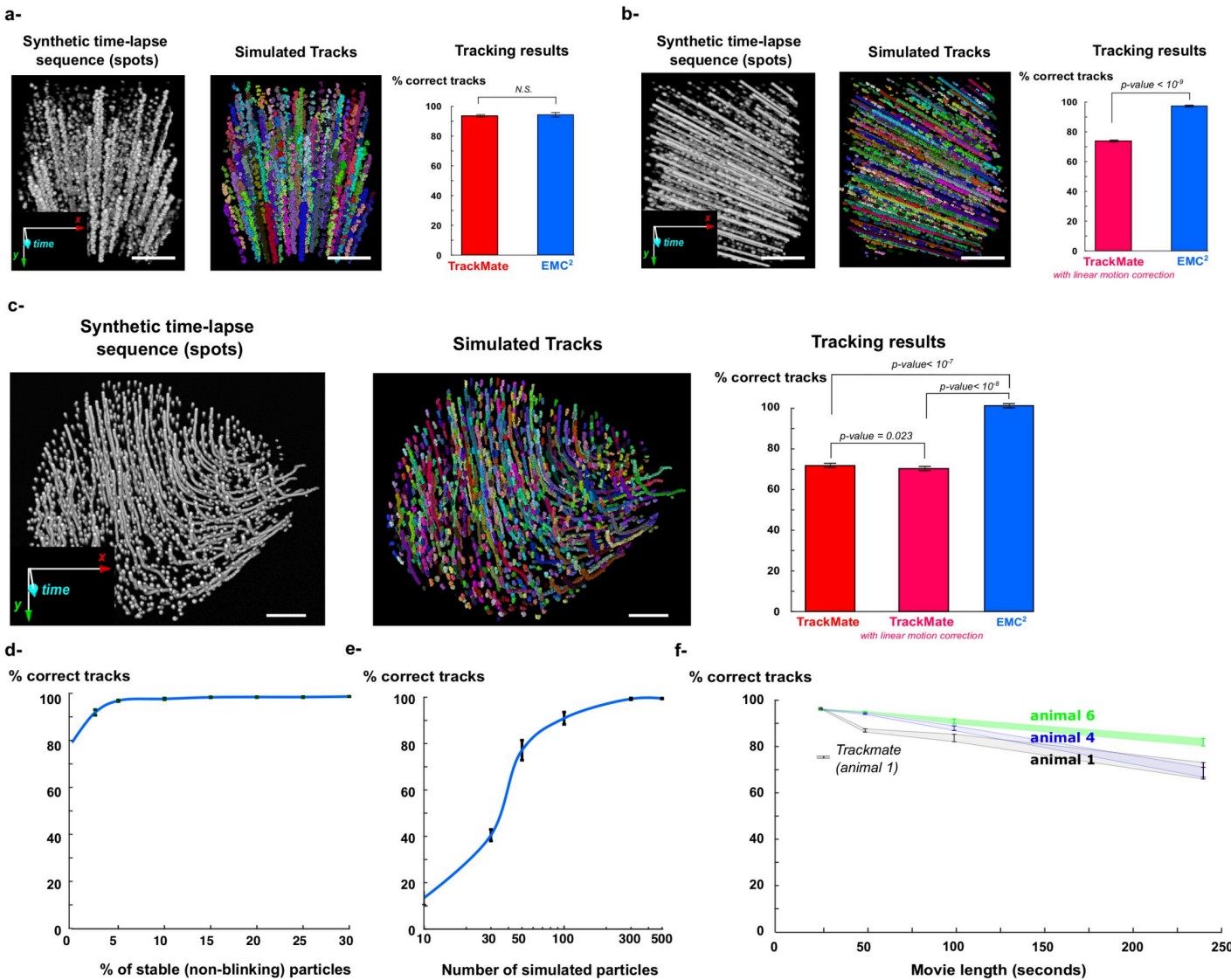

**Fig 3. Testing EMC² robustness with synthetic simulations.** For each simulated type of motion (confined diffusion (**a**), linear motion (**b**) and "Hydra-like" elastic deformation (**c**)), we simulated the stochastic firing of neuronal ensembles and corresponding fluorescence dynamics in synthetic time-lapse sequences (see Materials and Methods for details). We then compared the performances of EMC² (blue) with TrackMate (no motion correction (red) or linear motion correction (magenta)). P-values are obtained with the Wilcoxon rank sum test over n = 10 simulations in each case. (**d-e**) Using "Hydra-like" synthetic deformation, we measured the accuracy of EMC² for increasing proportion of stable (i.e. non-blinking) particles (neuronal cells) and increasing number of simulated particles. (**f**) After having estimated the deformation-field in three different animals (animal 1 (black), 4 (blue) and 6 (green)), we measured the accuracy of EMC² for simulated sequences with increasing length (25, 50, 100 and 240 seconds. Imaging and simulations were performed at 10 Hz). For comparison purposes, the performance of TrackMate algorithm for 25 seconds (animal 1), extracted from (**c**), is shown.

reconstructed tracks) linear motion correction, and that both were outperformed by EMC² (98.6 ± 0.3%, or 493.4 ± 1.6 correct tracks over 500.4 ± 0.2 reconstructed tracks).

We also measured the robustness of EMC² to parameter change in synthetic motion simulations. In particular, we measured the performance of the algorithm for an increased percentage of stable cells (Fig 3D) (see Material and Methods), an increased number of neurons (Fig 3E) and an increased length of simulated sequences (Fig 3F and Table 3). First, we found that even without stable cells ($\alpha_{stable}$ = 0), the accuracy of EMC² remained high (77.7% ± 3.6% (398.8 ± 18.4 correct tracks over 513.3 ± 1.7 reconstructed tracks), and rapidly increased to

**Table 3. Results of synthetic simulations (Hydra-like deformation, increasing length).**

| | 250 frames (25 s.) | | 500 frames (50 s.) | | 1000 frames (100 s.) | | 2400 frames (240 s.) | |
|---|---|---|---|---|---|---|---|---|
| | *Accuracy* | *Nb. Tracks* | *Accuracy* | *Nb. Tracks* | *Accuracy* | *Nb. Tracks* | *Accuracy* | *Nb. Tracks* |
| **Animal 1** | 98.6±0.3% | 500.4±0.2 | 87.5±0.8% | 511.7±1.0 | 83.8±1.8% | 512.0±3.1 | 67.4±4.1% | 538.8±8.0 |
| **Animal 4** | 98.0±0.2% | 503.5±0.4 | 95.9±0.4% | 507.9±1.1 | 88.7±1.2% | 521.5±2.6 | 66.7±2.5% | 571.9±8.5 |
| **Animal 6** | 97.8±0.2% | 502.9±0.7 | 96.9±0.3% | 505.6±0.7 | 92.0±1.3% | 515.7±2.3 | 81.7±1.9% | 540.7±5.4 |

91.8% ± 1.2% for $\alpha_{stable}$ = 5%, before reaching a plateau above 95% accuracy for $\alpha_{stable} \geq 10\%$. Conversely, we found that EMC$^2$ was very sensitive to the number of neurons, with poor performance for very few neurons (for simulations with only 10 neurons, the accuracy was 11.6% ± 0.6%, or 5.6 ± 0.3 correct tracks over 49.8 ± 0.8 reconstructed tracks). The accuracy rapidly increased to > 90% when more than 100 neurons were simulated. Tracking errors in simulations with few neurons are due to the inaccurate estimation of the deformation field, and iterative error propagation, when few fiducial *source* and *target* points are used. Finally, we measured how the performance of the EMC$^2$ evolved with the length of the simulated sequence, for three different animals (animals 1, 4 and 6, see Table 4). We chose animals with a high and homogeneous density of neurons so that we could accurately estimate, and therefore simulate, their body deformation. We found that the EMC$^2$ accuracy decreased with the sequence length, but remained high (*i.e.* >80%) for up to 2 to 4 minutes of simulation at 10 Hz, depending on the animal (animal 1 ~ 1200 frames = 2 min., animal 4 ~ 1500 frames = 2 min. 30s and animal 6 ~ 2400 frames = 4 min.). For longer simulations, the accuracy dropped below 80% and reached a mean of 72% at the end of the simulation (4 minutes). The number of reconstructed tracks remained close to 500 even for longer sequences (Fig 3F and Table 3). The decreased performance of EMC$^2$ for longer time-lapse sequences is an expected drawback of SPT-based tracking algorithms. Indeed, the probability of false associations between newly detected particles and existing tracks increases with time. Yet, the EMC$^2$ tracking algorithm, in contrast with state-of-the-art SPT algorithms, allows the robust, automatic tracking of individual neurons over few (~2–3) minutes in the highly deforming Hydra model. This sustained performance is particularly desirable as it allows the robust analysis of single neuron activity and functional coupling during different animal behaviors (section 4).

**Table 4. Results of statistical extraction of neuronal ensembles in Hydra (n = 8 animals).**

| *Animal* | *Movie length (frames)* | *Tracks (>150 frames)* | *Activity peaks* | *Ensembles* | *CB neurons* | *RP1 neurons* | *RP2 neurons* |
|---|---|---|---|---|---|---|---|
| *1* | *900* | *411* | *11* | *2* | | *139 (33.8%)* | *37 (9.0%)* |
| *1* | *1636* | *723* | *37* | *3* | *202(27.9%)* | *121(16.7%)* | *49(6.8%)* |
| *1* | *1371* | *607* | *27* | *3* | *222(36.6%)* | *78(12.9%)* | *41(6.8%)* |
| *1* | *988* | *516* | *23* | *3* | *220(42.6%)* | *49(9.5%)* | *55(10.7%)* |
| *2* | *1000* | *531* | *17* | *2* | *34(6.4%)* | *29(5.5%)* | |
| *2* | *1000* | *518* | *17* | *2* | *59(11.4%)* | *29(5.6%)* | |
| *3* | *1000* | *191* | *20* | *2* | *16(8.4%)* | *11(5.8%)* | |
| *4* | *1000* | *243* | *13* | *2* | *16(6.6%)* | *17(7.0%)* | |
| *4* | *1000* | *173* | *16* | *2* | | *24(13.9%)* | *8(4.6%)* |
| *5* | *1000* | *147* | *15* | *2* | *14(9.5%)* | *7(4.8%)* | |
| *6* | *1000* | *379* | *11* | *2* | *14(3.7%)* | *10(2.6%)* | |
| *7* | *1000* | *173* | *9* | *2* | *59(34.1%)* | *25(14.5%)* | |
| *8* | *1000* | *244* | *10* | *2* | *44(18.0%)* | *18(7.4%)* | |
| **Mean** | **1067** | **374** | **17.3** | | **82(18.7%)** | **43(10.8%)** | **38(7.6%)** |
| *Standard error* | *56* | *53* | *2.2* | | *26(4.4%)* | *12(2.3%)* | *8(1.8%)* |

Altogether these simulations show that EMC$^2$ is a robust tracking algorithm for different types of particle motions, and is therefore a versatile method for single particle tracking.

## 5 Tracking and analyzing single neuron activity in behaving animals

**a. Monitoring neuron activity with two-photon microscopy in mouse cortex.**    Two-photon calcium imaging is widely used to monitor single neuron activity in targeted brain regions of awake and behaving animals [3]. The head of the animal is usually fixed under the objective of the microscope, which limits the motion of neurons and facilitates their individual tracking. However, residual neuron motion due to animal movements, breathing or heartbeats requires computational post-processing of acquired time-lapse sequences to robustly monitor calcium activity of single neurons. The most popular technique for (slight) motion correction in time-lapse sequences is the elastic registration of fluorescence images with respect to one (or multiple) reference frame(s) using image intensity [13]. The computational load of image registration is important, especially for long sequences with large images, and algorithms have been developed to speed up the registration process and decrease the computational time to a few minutes for ~2,000 frame time-lapse sequences (with ~256x256 images) [14]. After image registration, the segmentation of neuronal masks for calcium fluorescence monitoring is then performed with standard intensity thresholding [38] or more elaborate techniques when neuronal masks are overlapping, such as non-negative matrix factorization [17].

To simultaneously localize and correct for neuron motion, we applied the EMC$^2$ Icy tracking protocol (Fig 2) to two-photon time-lapse calcium images of mouse visual cortex (Fig 4 and Material and Methods). Imaging was performed for 5 minutes at 12.3 Hz (~3,700 frame sequences) at days 1, 2 and 46. The entire EMC$^2$ protocol with neuron spot detection and tracking for each ~3,700 frame sequence ran in ~3 minutes with a 2.7 GHz Intel Core i7 processor. Consistent with [38], we observed a significant turn-over of active neurons (Fig 4A) with few neurons (median = 22% (15/68 neurons), n = 4 animals) that remained active across all days. We observed a similar number of active neurons on day 1 and 2, but a decreased number at day 46 which is probably due to several factors, such as decreased transgene expression or repeated experimental procedures [38]. We then analyzed single neuron trajectories obtained with EMC$^2$ (Fig 4B). Even if the animal's head was fixed under the microscope, residual motion of the field-of-view led to confined stochastic trajectories for single neurons. The positions of single neurons at each time were either computed with the intensity center of detected spots when neurons were detectable, or estimated using the computed elastic deformation of the field of view when neurons were silent and undetectable. We measured a median neuron displacement between frames of ~ 0.25 pixels, and a median maximum distance of excursion (relative to the center point of the trajectory) of ~ 4 pixels (Fig 4B).

Altogether, these results show that EMC$^2$ tracking protocol is a robust and fast method to post-process two-photon calcium imaging from awake mice. Trajectory analysis revealed the stochastic confined motion of single neuron positions, even in head-fixed animals. This residual motion is partly due to the animal's movements, but also to the uncertainty of sub-pixel localization of neuron spots at each time frame [39]. Moreover, the analysis of single neuron activity across days in the same cortical region showed a significant turn-over within the pool of active neurons each day, with few neurons remaining active over all days. Statistical analysis recently showed that this latter pool of active neurons could be a stable neuronal ensemble [38].

**b. Characterizing neuronal ensembles in behaving *Hydra*.**    There is increasing experimental evidence that neurons are organized into neuronal ensembles composed of a few tens of highly coupled neurons, and that these co-active ensembles are the fundamental

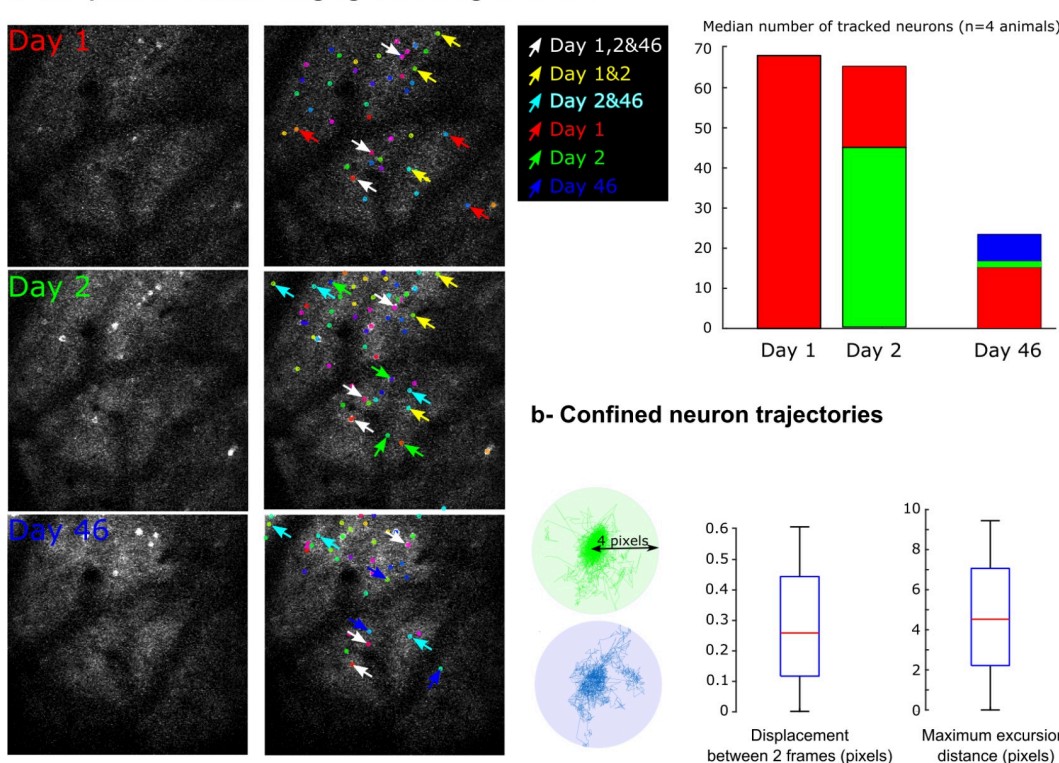

**Fig 4. Monitoring the activity of individual neurons in two-photon calcium imaging of mouse visual cortex with EMC².**
**a-** Two-photon calcium imaging of single neuron activity in visual cortex of awake mice is performed at Day 1, Day 2 and Day 46 during 5 minutes at 12.3 Hz. Tracking of neuron positions is performed with EMC² and reveals an important turn-over of active neurons across days. Examples of neurons that are active at Day 1, Day 2 or Day 46 are respectively highlighted with red, green or blue arrows. Neurons active at Day 1&2 are highlighted with yellow arrows, at Day 2&46 with cyan arrows, and at Day 1,2&46 with white arrows. The median number of active neurons each day is also plotted (n = 4 animals). The number of neurons that are active from Day 1, 2 or 3 are respectively represented in red, green or blue. **b-** Single neuron trajectories can be modeled with confined stochastic motion. Two example trajectories are shown (green & blue trajectories) with a maximum excursion distance of ~ 4 pixels. Boxplots of single neuron displacement between two consecutive frames, and maximum excursion distance (in pixels) are plotted (n = 590 trajectories).

computational units of the brain rather than single neurons themselves [21, 40]. Using manual annotation, it has been shown that *Hydra's* nervous system, one of the simplest of the animal kingdom, may be dominated by three main functional networks that are distributed through the entire animal [5, 41]. To confirm (or refute) these observations, we used EMC² and automatically tracked single neurons in *n = 13* time-lapse sequences (length 1067±56 frames at 10 Hz (Materials and Methods) from 8 different animals (Table 4 and Fig 5). Movies used in this study (avi and tiff files) can be downloaded from the BioStudies platform (https://www.ebi.ac.uk/biostudies/studies/S-BSST428). Analyzed movies were significantly longer than the manually annotated one (length = 200 frames at 10 Hz [5]). Ensemble activity, corresponding to the co-firing of neurons, can be detected as significant peaks within the raster plot of single neuron activity (Fig 5 and Materials and Methods). We detected a mean number of 1.7±2.2 peaks per movie, corresponding to a mean rate of 1 activity peak every 63 frames (6.3 s) which corresponds well with the 4 peaks observed previously in the 200 frame movie [5]. To associate each peak with a putative neuronal ensemble, we adopted a similar approach as in [42] and measured the similarity between these events in terms of the identities of the participating neurons (Fig 5 and Materials and Methods). We then performed k-means clustering of peak similarity,

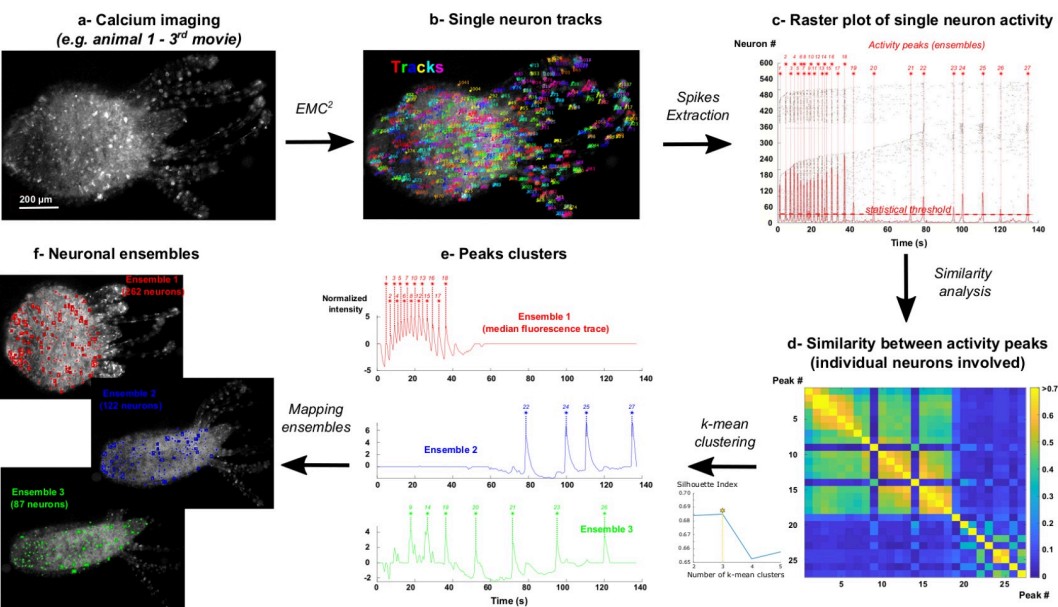

**Fig 5. Neuron tracking and mapping of neuronal ensembles in behaving Hydra. a-** Calcium imaging of single neuron activity in behaving Hydra. The images and analysis of the 3<sup>rd</sup> movie of animal 1 are given as representative examples. **b-** Single neuron tracks and fluorescence intensity are obtained with $EMC^2$ algorithm. **c-** For each neuron, spikes are extracted from fluorescence traces. Peaks of activity (highlighted with red stars) correspond to significant co-activity of individual neurons (sum of individual activities (solid red line) > statistical threshold (dashed red line), p = 0.001 see Materials and Methods). Each peak putatively corresponds to the activation of one neuronal ensemble. **d-** Similarity between activity peaks is computed using the identities of individual neurons that are firing at each peak (see Materials and Methods). **e-** The optimal number of peak classes (that putatively corresponds to the number of neuronal ensembles) is computed using the Silhouette index on k-means clustering of the similarity matrix (see Material and Methods). Median fluorescence trace of each neuronal ensemble and corresponding activity peaks are shown. The classification of individual neurons in each ensemble is determined based on their firing at ensemble peaks (see Materials and Methods). **f-** Individual neurons of each ensemble can be dynamically mapped in the original time-lapse sequence.

and used the Silhouette criterion [43] to determine the most likely number of neuronal ensembles causing the detected peaks of activity. We found 2 or 3 neuronal ensembles in each movie (3 neuronal ensembles were detected in 3 out of the 13 movies, or 23%). We then categorized each detected ensemble into one of the previously defined ensembles [5, 41]: Contraction Burst (CB) neurons that fire during longitudinal contraction of the animal, Rhythmic Potential 1 (RP1) that fire during the longitudinal elongation of the animal, and Rhythmic Potential 2 (RP2) neurons that fire independently of RP1 and CB activity. We found CB neuronal ensembles in almost all movies (11/13, or 85% of movies) and all animals (8/8, 100%), RP1 ensembles in all movies and animals and RP2 ensembles in fewer movies (5/13, or 38%) and only 2/8 (25%) animals. The absence of detected CB ensembles in 2 movies corresponds to the observed absence of contraction cycles within these movies. On the other hand, we hypothesize that the absence of RP2 ensembles in 6/8 (75%) of the animals is due to the limited depth of the field-of-view in confocal microscopy (see Materials and Methods). Indeed, RP2 neurons lie in the thin ectoderm of the animal [5] that may not have been imaged in some animals. Finally, we classified and mapped each individual neuron in the detected ensembles (see Materials and Methods). As in [5], we found that the CB ensemble was the most important group of neurons with a mean number of 82±26 neurons representing 18.7%±4.4% of the total number of neurons (a mean of 374±53 neurons were tracked over >150 frames in the different movies). The RP1 ensemble, with 43±12 neurons (10.8%±2.3%), was the next largest ensemble and RP2, with 38±8 (7.6%±1.8%), was the third. The relative number of neurons in the different

ensembles is in agreement with previous observations [5]. However, the overall size of each ensemble is smaller than the size reported previously. This is due to the fact that automatic classification of each individual neuron in an ensemble is more stringent than the manual classification that had been previously performed. Finally, we measured the overlap between ensembles, i.e. the proportion of single neurons that belonged to more than one ensemble. For each pair of ensembles, we computed the ratio between the number of shared neurons and the total number of neurons in both ensembles. We obtained ratios of 2.1%±0.7%, 2.4%±0.4% and 1.6%±0.6% respectively for CB-RP1, CB-RP2 and RP1-RP2 ensemble overlap. These very low values confirm the near non-overlap of the main neuronal ensembles in Hydra [5]. By coupling the automatic tracking of individual neurons in multiple *Hydra* with robust clustering analysis of neuronal activity, we were therefore able to confirm the previous observations that the *Hydra* nervous system is dominated by three main non-overlapping ensembles that are involved in different animal behaviors.

Altogether, these results show that EMC$^2$ is sufficiently robust to monitor single neuron activity in behaving animals. This constitutes a fundamental prerequisite for the analysis of neurons' functional organization and, ultimately, for our understanding of their emergent computational properties.

## Discussion

When using calcium imaging in living animals, an important challenge is the sustained tracking of neuronal positions over extended times in a moving, and potentially also deformable, environment. To tackle this issue, we have developed an algorithm, EMC$^2$, that tracks detectable particles with a state-of-the-art probabilistic tracking algorithm and uses the information contained in reconstructed tracks about the local deformation of the field-of-view (e.g. animal) to estimate the position of undetectable particles and potentially close tracking gaps. We validated the performance and versatility of EMC$^2$ by comparing its performance with a state-of-the-art tracking algorithm on manually tracked neurons in time-lapse calcium-imaging of behaving animals, including imaging over days of the same neurons from mouse visual cortex, the challenging deformable *Hydra*, and also on synthetic time-lapse sequences that modeled different types of motion/deformation of the field-of-view (confined diffusion, linear motion and *Hydra*-like elastic deformation). In all cases, EMC$^2$ showed high accuracy and outperformed state-of-the-art tracking methods.

Compared to traditional tracking approaches, composed only of particle detection and linking (see Table 1), our hybrid algorithm is better equipped to handle tracking gaps and general particle motion. Recently, tracking methods based on artificial neural networks [44, 45] have been introduced to handle more general particle motion, not only diffusion and/or linear motion. However, these methods are designed to link two sets of particle detections in consecutive time frames and, even if they can handle some missing or false detections, they do not appear well suited for tracking gaps such as those encountered in calcium imaging of neuronal activity.

On-going development of high-speed three-dimensional microscopes enables the imaging of neuronal activity with high temporal and spatial resolution in an increasing number of animal models [6, 9]. In the near future, the extension of EMC$^2$ tracking method to three-dimensional movies should not present important technical issues. Indeed, the thin-plate-spline transform used here to estimate the deformation of the field-of-view in two dimensions is actually a special case of polyharmonic splines that have been specifically designed for robust interpolation between data points in any dimensional space[46].

After having implemented EMC$^2$ in the open-source bio-imaging platform Icy [20], we tracked single neuron activity in behaving mouse visual cortex and also *Hydra*. In mouse cortex, our algorithm performed essentially flawlessly, and reveals the presence of neurons that continue to respond to the same stimulus over several days. In addition, using statistical clustering, we confirmed the previous observations that neural activity of *Hydra* is dominated by three major, non-overlapping neuronal ensembles that are involved in the animal's repetitive contraction and elongation. At the same time, automatic clustering of neuronal activity was not able to extract other smaller ensembles of the animal's nervous system, such as the tentacle and sub-tentacle ensembles [5] that we could observe by eye. Indeed, the coordinated activity of small ensembles is difficult to detect from individual neuronal spiking because of sparse activity and noise. Moreover, the activation of these smaller ensembles is less frequent than the activation of the three major ensembles and happened in only few movies. Therefore, the complete mapping of neuronal ensembles of *Hydra*, with the characterization of even the smallest neuronal ensembles with less frequent activity, will require in the near future further development of better imaging and tracking of single neuron activity over longer periods of time. For tracking neurons over longer time (several minutes or even hours), hardware implementations such as dual-color imaging with calcium-insensitive dye or partial immobilization of the animal [47] will be required. Another possible software strategy for increasing the robustness of EMC$^2$ tracking over long times would be the characterization and use of stereotypical poses of the animal [8] as reference frames for mapping the (almost) stable positions of neuronal subsets.

To conclude, our results show that EMC$^2$ is a robust and versatile tracking algorithm that allows monitoring and quantification of single neuron activity in behaving animals. Robust tracking of neural activity is a first step towards a better understanding of the neural code, i.*e.* how connected neuronal ensembles integrate information, underlie adaptive behavior and, more generally, compute the animal's behavioral or internal states.

## Materials and methods

### 1 Elastic motion correction and concatenation (EMC$^2$) of short tracks

The eMHT algorithm returns a set of $N$ short tracks (tracklets) $\tau_i$, for $1 \leq i \leq N$, of detectable particles; the $i^{\text{th}}$ tracklet $\tau_i$ starts at position $\boldsymbol{x}_i(t_i^s)$ at time $t_i^s$ and ends at position $\boldsymbol{x}_i(t_i^e)$ at time $t_i^e$. To concatenate tracklets, i.e. to link the tracklets that putatively correspond to the same neuron, we then estimate the backward position $\hat{\boldsymbol{x}}_i^b(t)$ of the particle (neuron) corresponding to the $i^{th}$ tracklet before the starting-point ($t \leq t_i^s$) of the tracklet. To estimate $\hat{\boldsymbol{x}}_i^b(t)$ for $t \leq t_i^s$, we iteratively apply backward thin-plate-spline transformation $\hat{\boldsymbol{x}}_i^b(t-1) = TPS_{backward}(\hat{\boldsymbol{x}}_i^b(t))$, with the initial condition $\hat{\boldsymbol{x}}_i^b(t_i^s) = \boldsymbol{x}_i(t_i^s)$. Similarly, to estimate the forward position $\hat{\boldsymbol{x}}_i^f(t)$ after the ending-point of the tracklet, i.e. for $t \geq t_i^e$, we iteratively apply the forward transformation $\hat{\boldsymbol{x}}_i^f(t) = TPS_{forward}(\hat{\boldsymbol{x}}_i^f(t-1))$ with the initial condition $\hat{\boldsymbol{x}}_i^f(t_i^e) = \boldsymbol{x}_i(t_i^e)$. From the estimated positions of particles corresponding to each tracklet, we then compute the distance $d_{i,j}$ between tracklets $\tau_i \neq \tau_j$ with

$$d_{i,j} = \begin{cases} \infty, \text{ if } t_i^e > t_j^s \text{ or } t_j^e > t_i^s \\ \min_{t_i^e \leq t \leq t_j^s} \|[\hat{\boldsymbol{x}}_i^f - \hat{\boldsymbol{x}}_j^b](t)\|, \text{ if } t_i^e \leq t_j^s \leq t_i^e + gap_{max} \\ \min_{t_j^e \leq t \leq t_i^s} \|[\hat{\boldsymbol{x}}_j^f - \hat{\boldsymbol{x}}_i^b](t)\|, \text{ if } t_j^e \leq t_i^s \leq t_j^e + gap_{max} \end{cases}$$

where $gap_{max}$ is a user-defined maximum time gap that EMC$^2$ is allowed to close (typically a few hundreds of frames). We need to apply a maximum time gap for time-lapse sequences

with particles that remain undetectable over long periods of time due to the error growth during backward/forward estimation of the putative position of undetectable particles.

Using the computed distances between tracklets, we then define an association cost matrix,

$$\mathbf{\Phi} = \begin{pmatrix} \phi_{1,1} & \cdots & \phi_{1,N} \\ \vdots & \ddots & \vdots \\ \phi_{N,1} & \cdots & \phi_{N,N} \end{pmatrix},$$

with $\phi_{i,j} = d_{i,j}$ if $d_{i,j} < d_{max}$, and $\phi_{i,j} = \infty$ otherwise. $d_{max}$ is the second user-defined parameter of our tracking algorithm that specifies the maximum distance allowed between the forward-propagated end-point of a tracklet and the backward-propagated starting-point of another tracklet. Finally, among all possible associations for which the cost $\phi_{i,j} < \infty$, the optimum set of concatenated tracklets $\tau_{i^*} \rightarrow \tau_{j^*}$ among all the tracklets $\{\tau_i, \tau_j\}$, $1 \leq i,j \leq N$ is the solution of the global minimization problem

$$i^* \rightarrow j^* = \min_{i,j \; such \; that \; \phi_{i,j} < \infty} \sum \phi_{i,j}.$$

This minimization problem, known as the assignment problem, is similar to the problem solved in GDM methods of tracking, where algorithms determine the optimal association between particle detections by minimizing the global distance between detections in consecutive time frames of the sequence. One of the first and most popular algorithms to solve assignment problems is the Hungarian algorithm [48]. However, due to its computational load, faster algorithms have been proposed over the years. We used here the Jonker-Volgenant algorithm [49] implemented in the *TrackMate* plugin in ImageJ (see Table 1).

Finally each particle (neuron) track $T_i$ for $1 \leq i \leq N'$, with $N' \leq N$ the number of "long" tracks, results from the concatenation of $n_i$ tracklets: $T_i = \tau_{i_1} \rightarrow \tau_{i_2} \rightarrow \cdots \rightarrow \tau_{i_{n_i}}$ with $t^e_{i_1} \leq t^s_{i_2} \leq \cdots \leq t^s_{i_{n_i}}$. We highlight that, by construction, each long track $T_i$ does not necessarily span the whole time-lapse imaging sequence, but begins at the starting-time $t^s_{i_1}$ of the first tracklet $\tau_i$ and ends at the ending-time $t^e_{i_{n_i}}$ of the last tracklet $\tau_{i_{n_i}}$.

## 2 Icy protocol

**a Detection of spots (e.g. neurons) in time-lapse sequences.** To detect automatically the positions of fluorescent spots, corresponding to detectable particles, in each frame of the time lapse sequence, we designed a multi-step algorithm (see Fig 2) where we first detected fluorescent spots that are significantly brighter than background with a fast and robust algorithm based on a wavelet transformation of the image and statistical thresholding of the wavelet coefficients (*block* number 1 in Icy protocol (Fig 2)) [50]. These spots correspond to individual particles or clusters of particles (e.g. neurons). To separate individual particles in the detected clusters, we then multiplied the original sequence with the binary mask obtained with wavelet thresholding and convolved the result of the multiplication with a log-Gaussian transformation (*block* number 2). The log-Gaussian convolution is similar to the point-spread function of microscopes and thus enhances individual particles [51]. Finally, we extracted the positions of single particles by applying a local-maxima algorithm (*block* number 3) to the convolved sequence.

**b Tracking (EMC$^2$).** After having detected the positions of fluorescent spots in each time frame, a second series of *blocks* computed the tracks of each single particle. First, *block* number 4 used the positions of spots and a robust Bayesian algorithm (eMHT [30]) to compute single

tracks of detectable particles. Due to fluctuating detectability, many computed tracks are terminated prematurely and new tracks are created when particles are detectable again. We thus applied the EMC² algorithm (block number 5) to close gaps and reconstruct single-particle tracks over the entire time lapse sequence.

The tracking protocol can be found here: http://icy.bioimageanalysis.org/protocol/detection-with-cluster-un-mixing-and-tracking-of-neurons-with-emc2/ and is also directly accessible through the search bar of the Icy software. A step-by-step tutorial for tracking neurons and exporting track intensity with Icy is provided as Supplementary material.

## 3 Validation of EMC²

**a Comparison metric.**    To compare the tracks obtained with EMC² and other automatic tracking algorithms with *ground truth* tracks, we first considered the whole set of detections $x_i(t)$, $1 \leq i \leq N(t)$, with $N(t)$ the number of detections at time $1 \leq t \leq T$ ($T$ being the length of the time sequence) and assigned each detection to the closest active track at time $t$. Therefore, for each reference track $\theta_j^r$, $1 \leq j \leq |\Theta^r|$, with $|\Theta^r|$ the total number of reference tracks, and for each test track $\theta_k^t$, $1 \leq k \leq |\Theta^t|$, with $|\Theta^t|$ the total number of test tracks, we obtained a set of associated detections. We then considered that a test track matched a reference track if it shared at least 80% of common detections. Finally, for each reference track, we either obtained no test track that matched, exactly one test track that matched or more than one.

**b Synthetic motions.**    To validate the robustness and accuracy of EMC² in different scenarios, we simulated three classes of motions: **confined diffusion, linear motion and elastic deformation.** For **confined diffusion**, each simulated spot (e.g. fluorescent neuron) can diffuse with coefficient $D = 1$ *pixel² per frame* and is confined to a 10 pixel disk area. For **linear motion**, each simulated spot moves linearly at speed $v = 1$ *pixel per frame*. When a track reaches the boundary of the field-of-view (a rectangle of 200x200 pixels), it is terminated and another track is initiated at the other side of the FOV. Finally, for **elastic deformation**, we used the experimental tracks in *Hydra* to estimate iteratively (i.e. from one frame to the following one) the local deformation for each synthetic track position.

**c Firing rates of individual neurons.**    To model the stochastic firing rates of individual neurons (total number of neurons $n_{neurons}$), we first determined a proportion ($\alpha_{stable}$) of stable spots, i.e. non-blinking cells, with constant fluorescent intensity. In *Hydra*, stable cells typically correspond to nematocytes or other cell types that also express fluorescent proteins after the genetic editing of the animal, but that don't fire as neurons do [5]. To simulate the correlated activity patterns observed in *Hydra*, we then divided the $(1-\alpha_{stable})n_{neurons}$ firing neurons, with intermittent activity and detectability, into $n_{group}$ ensembles. All neurons in each ensemble fire simultaneously with Poisson rate $\lambda_{group} = size_{group} \lambda_{individual}$, with $\lambda_{individual}$ the firing rate of individual neurons and $size_{group} = (1-\alpha_{stable})n_{neurons}/n_{group}$ the number of neurons in each group. Parameters for each simulation used for the validation of EMC² algorithm are summarized in Table 5.

**d Generation of synthetic images.**    To generate synthetic fluorescence time-lapse sequences, we used a mixed Poisson-Gaussian model [31]. In this model, the intensity $I[x,y]$ at pixel location $[x,y]$ is equal to $I[x,y] = U[x,y]+N(0,\sigma_n)$ where $U$ is a random Poisson variable and $N(0,\sigma_n)$ is additive white Gaussian noise with standard deviation $\sigma_n$. The intensity $\lambda[x,y]$ of the Poisson variable varies spatially because it depends on the presence or not of particle spots (neurons). Therefore, $\lambda[x,y] = P[x,y]+B$ with $B$ a constant background value and $P[x,y]$ the spots' intensity at position $[x,y]$. Assuming an additive model for the intensity of the spots, $P[x, y] = \sum_{i=1..n_{neurons}} P_i[x, y]$, where $P_i[x,y]$ is the signal originating from the $i^{th}$ spot. We approximated the point-spread-function (PSF) of the microscope with a Gaussian profile.

**Table 5. Parameters used for synthetic simulations.**

| Parameters | $n_{neuron}$ | $\alpha_{stable}$ | $n_{group}$ | $\lambda_{individual}$ | $A$ | $\tau_{decay}$ | $\beta$ | $\mu$ | $\tau_{rise}$ | $\sigma_{PSF}$ | $\sigma_n$ | $B$ | $SNR = \frac{A}{B+\sigma_n^2}$ |
|---|---|---|---|---|---|---|---|---|---|---|---|---|---|
| Name | Total number of neurons | % of non-firing spots | Number of neuron groups | Individual firing rate | Amplitude | Fluorescence decay time constant | Decay power | Median rising time | Rising time constant | St. Dev. of the PSF | St. Dev. of the Gaussian noise | Poisson background | Signal-to-noise ratio |
| **Confined diffusion & Linear motion** | 150 | 20% | 10 | 0.01 frame$^{-1}$ | 100 | 3 frames | 1 | 1 frame | 0.5 frames | 1 pixel | 5 | 10 | $\approx 3$ |
| **Elastic deformation (Hydra)** | 500 | 20% | 10 | 0.0002 frame$^{-1}$ | 100 | 15 frames | 2 | 2 frames | 0.5 frames | 1 pixel | 5 | 10 | $\approx 3$ |

Thus, for a particle located at position $[x_0^i, y_0^i]$, its intensity at position $[x,y]$ is given by

$P_i[x, y] = A_i \exp -\frac{(x-x_0^i)^2 + (y-y_0^i)^2}{2\sigma_{PSF}^2}$, with $A_i$ the amplitude of the $i^{th}$ particle and $\sigma_{PSF}$ the standard deviation of the 2D Gaussian profile of the PSF. We chose a constant amplitude for each particle $A = A_i$, for all $1 \leq i \leq n_{neurons}$. Parameters used in simulations are summarized in Table 5.

**e Fluorescence kinetics.** When a neuron fires at time $t_0$, we model its fluorescence time course with the general kinetics equation

$$f(t) = A \frac{\exp\left(-\left(\frac{t-t_0}{\tau_{decay}}\right)^\beta\right)}{1 + \exp\left(-\frac{t-t_0-\mu}{\tau_{rise}}\right)},$$

where the numerator models a power-law exponential decay of the fluorescence ($\beta = 1$ models a standard single exponential decay), with a decay time constant $\tau_{decay}$, and the denominator models a sigmoidal increase of fluorescence with median $\tau_{rise}$ and time constant $\mu$. Kinetic parameters for each synthetic simulation are summarized in Table 5. These parameters were obtained by fitting $n = 3075$ individual spikes from 444 individual neuron tracks in an experimental time-lapse sequence (250 frames at 10 Hz) of GCAMP-labeled *Hydra* [5] (S1 Fig). We highlight that, for *Hydra* elastic simulations, we used a long decay time constant and a power index $\beta = 2$ instead of 1 for standard confined diffusion and linear motion simulations.

## 4 Tracking and analyzing single neuron activity in living animals

**a Two-photon calcium imaging of mouse visual cortex.** Movies used in this study are issued from [38], and experimental protocol for two-photon volumetric imaging of targeted brain regions in mouse visual cortex can be found in the Methods section of this manuscript.

**b Hydra maintenance.** *Hydra* were cultured using standard methods [52] in Hydra medium at 18˚C in the dark. They were fed freshly hatched *Artemia nauplii* twice per week.

**c Hydra Imaging.** Transgenic *Hydra* expressing GCaMP6s in the interstitial cell lineage were used and prepared for imaging studies as previously described [5]. Calcium imaging was performed using a custom spinning disc confocal microscope (Solamere Yokogawa CSU-X1). Samples were illuminated with a 488 nm laser (Coherent OBIS) and emission light was detected with an ICCD camera (Stanford Photonics XR-MEGA10). Images were captured with a frame rate of 10 frames per second using either a 6X objective (Navitar HRPlanApo 6X/0.3) or a 10X objective (Olympus UMPlanFl 10x/0.30 W).

All movies used in this study can be downloaded from the BioStudies website https://www.ebi.ac.uk/biostudies/studies/S-BSST428.

**d Extracting single neuron activity.**    We extracted the fluorescence trace of each individual neuron using the *Track Processor Intensity profile* within the *TrackManager* plugin in Icy [20] (a step-by-step tutorial for tracking neurons and exporting track intensity with Icy can be found on the EMC[2] plugin web documentation: http://icy.bioimageanalysis.org/plugin/elastic-motion-correction-concatenation-emc2-of-tracks/). For each detection within the track, the extracted intensity corresponded to the mean intensity over a disk centered at the detection's position, with a 2 pixels diameter. When detections are missing (tracking gap), the intensity was set to 0. Then, for each individual neuron, we denoised its non-zero fluorescence trace using wavelet denoising (*wdenoise*) in Matlab. We then automatically extracted spikes with a custom procedure where we first computed the discrete derivative of the smoothed fluorescence signal, then set to 0 all negative variations and finally, we detected significant positive variations of the signal (discrete positive derivative > quantile at 98% of all empirical positive variations) that putatively corresponded to spikes.

**e Statistical characterization of neuronal ensembles in Hydra.**    Neuronal ensembles are groups of neurons that repeatedly fire together. Therefore, the activity of neuronal ensembles can be detected as significant co-activity peaks in the raster plot of single neuron firing. To detect significant peaks of activity, we applied the procedure described in [42], and identified as peaks times at which the sum of single neuron activity within a time step of 100 ms fell within the quantile at 0.999% obtained empirically by circularly shuffling the individual spikes in the activity raster plot.

Then, to relate detected peaks of activity to putative neuronal ensembles, we constructed a vector describing the activity of each individual neuron at the detected peaks with entries 1 if the neuron fires at that peak, and 0 otherwise. We then computed the similarity between these vectors for each of the activity peaks, using the Jaccard index:

$$Jaccard(peak\ i, peak\ j) = \frac{number\ of\ neurons\ that\ fire\ at\ peak\ i\ \textbf{and}\ peak\ j}{number\ of\ neurons\ that\ fire\ at\ peak\ i\ \textbf{or}\ peak\ j}.$$

Then, to estimate the number of neuronal ensembles underlying the detected peaks of activity, we clustered the peaks with a k-means algorithm based on their similarity for different numbers of classes (from 1 to 5 classes typically). K-means clustering was performed using the cosine distance. The optimal number of classes in the k-means clustering algorithms, and therefore the putative number of neuronal ensembles, was computed using the Silhouette index [43]. For a clustering of $N$ peaks into $k$ classes, the Silhouette index is given by $Silhouette(k) = \frac{1}{N}\sum_{i=1}^{N}(b_i - a_i)/\max(a_i, b_i)$ where $a_i$ is the average distance from the $i^{th}$ peak to the other peaks in the same cluster as $i$, and $b_i$ is the minimum average distance from the $i^{th}$ peak to peaks in a different cluster, minimized over clusters. An advantage of the Silhouette evaluation criterion over other clustering criteria is its versatility, as it can be used with any distance (cosine distance was used here for the k-means clustering). Finally, we assigned each individual neuron to an ensemble if that neuron fired in more than 50% of the activity peaks of the identified ensemble.

## Supporting information

**S1 Fig. Fitting the fluorescence kinetics of firing neurons in calcium imaging of Hydra.**
Fluorescence time course of a firing neuron (at time $t_0$) is modeled with the kinetics equation

$$f(t) = A\frac{\exp\left(-\left(\frac{t-t_0}{\tau_{decay}}\right)^{\beta}\right)}{1 + \exp\left(-\frac{t-t_0-\mu}{\tau_{rise}}\right)},$$

where the numerator models a power-law exponential decay of the fluorescence with a decay time constant $\tau_{decay}$, and the denominator models a sigmoidal increase of fluorescence with median $\tau_{rise}$ and time constant $\mu$. Boxplot for fitted parameters ($n$ = 3075 individual spikes from 444 individual neuron tracks) are shown.
(EPS)

**S2 Fig. Comparing tracking methods with manual ground truth on calcium imaging data in behaving Hydra.** Tracks obtained with TrackMate (red), EMC$^2$ without motion correction (purple) and EMC$^2$ are compared to manual ground truth obtained by manual concatenation of tracklets in calcium imaging of behaving Hydra (see Material and Methods).
(EPS)

## Acknowledgments

We thank John Wang for help, lab members, MBL staff and members of the MBL *Hydra* lab for support and advice.

## Author Contributions

**Conceptualization:** Thibault Lagache, Adrienne Fairhall, Rafael Yuste.

**Data curation:** Thibault Lagache, Alison Hanson, Jesús E. Pérez-Ortega.

**Formal analysis:** Thibault Lagache.

**Funding acquisition:** Rafael Yuste.

**Investigation:** Thibault Lagache, Rafael Yuste.

**Methodology:** Thibault Lagache.

**Project administration:** Rafael Yuste.

**Resources:** Alison Hanson, Jesús E. Pérez-Ortega.

**Software:** Thibault Lagache.

**Supervision:** Rafael Yuste.

**Validation:** Thibault Lagache.

**Writing – original draft:** Thibault Lagache.

**Writing – review & editing:** Thibault Lagache, Alison Hanson, Jesús E. Pérez-Ortega, Adrienne Fairhall, Rafael Yuste.

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
