## [Decision Letter · Decision Letter 0]

21 Oct 2020

Dear Dr. Lagache,

Thank you very much for submitting your manuscript "Robust tracking of single neuron calcium dynamics in behaving Hydra" for consideration at PLOS Computational Biology.

As with all papers reviewed by the journal, your manuscript was reviewed by members of the editorial board and by several independent reviewers. In light of the reviews (below this email), we would like to invite the resubmission of a significantly-revised version that takes into account the reviewers' comments.

We cannot make any decision about publication until we have seen the revised manuscript and your response to the reviewers' comments. Your revised manuscript is also likely to be sent to reviewers for further evaluation.

Sincerely,

Boris S. Gutkin

Associate Editor

PLOS Computational Biology

Lyle Graham

Deputy Editor

PLOS Computational Biology

Reviewer's Responses to Questions

**Comments to the Authors:**

Reviewer #1: The review is uploaded as a PDF attachment.

Reviewer #2: Lagache and colleagues present a complete user-friendly software package for tracking neurons during short calcium recordings in moving and deforming hydra. They demonstrate that their method can track neurons even when those neurons are inactive, without the need for an additional fluorescent marker. The development of effective algorithms that are integrated into user-friendly and open source software packages like Icy is very valuable for the community.

There are two substantive concerns.

1) It is unclear how the method performs on recordings longer than 25 seconds.

I could not find any description of the effective recording length for the validation performed against synthetic data with empirically derived noise, so I don't know if that was greater than 25s. But it is important to show that the method can work over longer recordings because calcium recordings are often many minutes long, and the sequential nature of this method would seem to be particularly susceptible to accumulating errors over time. For example, I might reasonably expect to accumulate errors in stitching tracks together such that after a few minutes a significant fraction of neurons may now be assigned different labels at the end of the recording compared to the beginning. One way the authors could address this concern is by showing how performance degrades over recording length when validated against realistic synthetic data.

2) Authors should clarify novelty and correct description of a previous method.

A key selling point of the method is that it uses deformation defined by tracked neurons to infer the location of missing neurons during gaps. The text highlights this method extensively, as if it were novel. But method [9], Ngueyn et al, explicitly performs the same step. Like this method, [9] uses thin plate splines of tracked neurons to infer the position of missing neurons, see main text accompanying Fig 5 in that reference [9].

The authors make a special point to explicitly compare to reference [9] in detail (for example on p. 7) and they also compare to many other methods in Table 1 and throughout the manuscript, all of which is commendable. However, they incorrectly claim that [9] is " designed to link two sets of particle detections in consecutive time frames." But this is incorrect. The method in [9] is time independent, in the sense that all the recorded volumes could be shuffled in time without changing tracking performance. Similarly, Table 1 incorrectly states that ref [9] is not robust to long gaps. On the contrary, because it is time independent, ref [9] is robust to exceptionally long gaps.

The authors should clarify novelty and correct descriptions of the previous method.

Additional technical concern:

There seems to be an inconsistency with Figure 4. It shows calcium activity lasting 1,400 s or 23 mins, whereas the main text and Table 3 suggests that recordings lasted less than 2 mins (1000 frames @ 10hz). Please resolve this inconsistency.

**Have all data underlying the figures and results presented in the manuscript been provided?**

Reviewer #1: Yes

Reviewer #2: None

PLOS authors have the option to publish the peer review history of their article (what does this mean?). If published, this will include your full peer review and any attached files.

Reviewer #1: No

Reviewer #2: No
---

## [Decision Letter · Decision Letter 1]

30 Mar 2021

Dear Dr. Lagache,

Thank you very much for submitting your manuscript "EMC2: A versatile algorithm for robust tracking of calcium dynamics from individual neurons in behaving animals" for consideration at PLOS Computational Biology.

As with all papers reviewed by the journal, your manuscript was reviewed by members of the editorial board and by independent reviewers. In light of the reviews (below this email), we would like to invite the resubmission of a significantly-revised version that takes into account the reviewers' comments.

In particular, it is necessary that you address the concerns of Reviewer #2 regarding the relevant time scale for the method, which is a critical factor for it's experimental application.

We cannot make any decision about publication until we have seen the revised manuscript and your response to the reviewers' comments. Your revised manuscript is also likely to be sent to reviewers for further evaluation.

Sincerely,

Boris Gutkin

Associate Editor

PLOS Computational Biology

Lyle Graham

Deputy Editor

PLOS Computational Biology

Reviewer's Responses to Questions

**Comments to the Authors:**

Reviewer #1: The authors have more than adequately adressed all my relevant comments. The addition of a second test case and the exploration of the performances of EMC2 with respect to the fraction of stable cells, cell density and time of recording both help in demonstrating the versatility of the method. I believe the paper will acquire greater visibility and applicability as it now stands. I also appreciated the clarifications the authors made on the stitching procedure both in the Methods section and in modifying Fig1. Finally, quantification of performances of the different methods in terms of reconstructed tracks is very instructive and allows for a rational comparison. I strongly support publication of this paper.

Reviewer #2: In this revision Lagache and colleagues have made important revisions and clarifications. It remains unclear, however, whether the method can track neurons in hydra over a sufficiently long timescale to be useful.

The authors specifically motivate their work by the need to "achieve robust and automatic tracking of individual cells over long time-lapse sequences." Most calcium activity experiments are for at least a few minutes, and recent published work by this group recorded calcium activity in moving hydra for 2 hours (Yamamoto and Yuste 2020, see Fig 3). It is therefore imperative that the authors show what accuracy to expect on the 5 min timescale and beyond.

With synthetic data, the authors now show that their method's accuracy drops from 97% for 250 frames (presumably 25s) to an accuracy of ~80% for 1000 frames (presumably 100s). It seems possible that accuracy will continue to drop dramatically as the timing approaches 5 mins.

Related: synthetic data is reported in terms of frames, but the authors should instead report the equivalent time in seconds, based on the timescale of the motion in the synthetic data. The authors should make this easy to find in the text and figure or caption.

If the method is unable to achieve reasonable accuracy on the multiple minutes timescale, the authors should scale back their claims, and it may also reduce the significance of the work.

**Have all data underlying the figures and results presented in the manuscript been provided?**

Reviewer #1: Yes

Reviewer #2: None

PLOS authors have the option to publish the peer review history of their article (what does this mean?). If published, this will include your full peer review and any attached files.

Reviewer #1: No

Reviewer #2: No
---

## [Decision Letter · Decision Letter 2]

13 May 2021

Dear Dr. Lagache,

Thank you very much for submitting your manuscript "EMC2: A versatile algorithm for robust tracking of calcium dynamics from individual neurons in behaving animals" for consideration at PLOS Computational Biology.

As with all papers reviewed by the journal, your manuscript was reviewed by members of the editorial board and by several independent reviewers. In light of the reviews (below this email), we would like to invite the resubmission of a significantly-revised version that takes into account the reviewers' comments. In particular the authors must address the issues brought up by reviewer 1 on the limitations of the method for extended recording times that had come up after the last revision. Indeed, in order to meet the editorial requirements of PLoS CB it is necessary to show that the method presents a significant advance on existing methods

We cannot make any decision about publication until we have seen the revised manuscript and your response to the reviewers' comments. Your revised manuscript is also likely to be sent to reviewers for further evaluation.

Sincerely,

Boris S. Gutkin

Associate Editor

PLOS Computational Biology

Lyle Graham

Deputy Editor

PLOS Computational Biology

Reviewer's Responses to Questions

**Comments to the Authors:**

Reviewer #1: I still support the publication of this paper.

Reviewer #2: With further simulation studies, the authors now show that their method's performance degrades as recording duration increases. For Hydra, the method is only suitable for very short recordings (<5 min timescale). This contradicts claims made in the abstract and throughout the text related to "long time lapse sequences" or "long-term" monitoring. Claims about closing potentially "long" gaps also lack evidence. In short there is a disconnect between many of the authors' claims and the evidence.

Evidence does not support the authors claim that that this method "greatly supersedes existing approaches." In contrast, other methods (NeRVE from Ngueyn et al. 2017; Conditional Random Fields from Chaudhary et al. elife 2021) are fundamentally time-independent and thus maintain the same performance regardless of recording duration. The authors need to revise their comparisons to other methods with a frank assessment of this limitation, which seems to arise from the sequential time-dependent nature of the algorithm. For example, it would be relevant to report in Table 1 whether each algorithm is time-dependent or not.

Crucially, the authors need to better justify why this method is meritorious despite being limited to only very short recordings. To be compelling the authors should provide specific use cases, and also include a discussion of why and how errors accumulate with time and what fundamentally about the algorithm imposes this limitation.

Minor, but important: it is imprecise to claim that accuracy "remained above ~70% for all animals" when Table 3 reports accuracy of 66.7% and 67.4% for two of the three animals.

**Have the authors made all data and (if applicable) computational code underlying the findings in their manuscript fully available?**

Reviewer #1: Yes

Reviewer #2: None

PLOS authors have the option to publish the peer review history of their article (what does this mean?). If published, this will include your full peer review and any attached files.

Reviewer #1: **Yes: **Olivier Cochet-Escartin

Reviewer #2: No
---

## [Decision Letter · Decision Letter 3]

7 Jul 2021

Dear Dr. Lagache,

Thank you very much for submitting your manuscript "EMC2: A versatile algorithm for robust tracking of calcium dynamics from individual neurons in behaving animals" for consideration at PLOS Computational Biology. As with all papers reviewed by the journal, your manuscript was reviewed by members of the editorial board and by several independent reviewers. The reviewers appreciated the attention to an important topic. Based on the reviews, we are likely to accept this manuscript for publication, providing that you modify the manuscript according to the review recommendations. Please follow the suggestions laid out by the reviewer 2.

Sincerely,

Boris S. Gutkin

Associate Editor

PLOS Computational Biology

Lyle Graham

Deputy Editor

PLOS Computational Biology

[LINK]

Reviewer's Responses to Questions

**Comments to the Authors:**

Reviewer #1: I thank the authors for the clarifications of time-dependent and time-independent methods following Reviewer 2's comments. I feel these clarifications now clearly define both the advantages and limitations of the EMC2 method in comparison with previously developped solutions. By clearly discussing in which situations, such as Hydra neuronal tracking, EMC2 provides improvement from existing methods, the authors now clearly provide the reader with both a novel tracking method and rational arguments for using it, or not, depending on her/his own experimental data.

Reviewer #2: The authors have taken important steps to bring their claims more into alignment with the evidence they present, but some problematic language remains.

• The phrase “Long time lapse sequences” still appears on p. 4 This should be removed because it implies that the method will address the “long time lapse” case, when it does not.

• The placement of new language about “…premature ending of a track due to missing detection(s), can lead to important error propagation” on p. 8 is problematic. That text appears in subsection “2 - Limitations of Standard SPT Algorithms,” but the authors claim incorrectly that EMC^2 overcomes those limitations of standard SPT algorithms in subsection “3 – EMC^2 Algorithm”: “To overcome these limitations, we have developed EMC2…” The authors need to revise this framing so that they do not incorrectly claim that they have overcome the limitations on accumulation of errors over time.

• The claim (p.16) that EMC^2 is robust “independent of the type and complexity of particle motion” is not supported by the data. Indeed if this were true it would be extraordinary. I suspect that the performance of EMC^2, like all time-dependent tracking algorithms before it, is dependent on the amount and type of motion per frame, and with enough motion per frame its performance would degrade precipitously. Please qualify this claim or provide evidence to support it.

Minor, but important:

• This statement on p. 7 incorrectly implies that the sparse detectability of neurons is uniquely problematic for Hydra and not C. elegans or other organisms, when in fact this depends only on whether the animal is expressing a reference fluorophore like RFP, and is not an inherent property of the organism. Please reword and clarify.

“while the aforementioned mapping methods used in C. Elegans can accommodate moderate changes in the total number of neurons between frames due to missing or spurious detections (counting noise), the intermittent and sparse detectability of neurons in calcium imaging definitely hinders the applicability of mapping methods to Hydra [7]”

• Subsection “4 - Validation of EMC^2”: “We used the first 250 frames” p.13 The authors should specify in the main text the duration in seconds of these 250 frames (Perhaps 25 seconds?) This information is important for the reader to interpret the method’s performance.

• In the abstract the authors write: “Our results prove…”. The statement of proof is unnecessarily, and in my opinion, inappropriately strong. Best usage would be to reserve proofs for mathematical proofs. Please revise.

**Have the authors made all data and (if applicable) computational code underlying the findings in their manuscript fully available?**

Reviewer #1: Yes

Reviewer #2: Yes

PLOS authors have the option to publish the peer review history of their article (what does this mean?). If published, this will include your full peer review and any attached files.

Reviewer #1: No

Reviewer #2: No

Figure Files:

Data Requirements:

Reproducibility:

References:

---

## [Decision Letter · Decision Letter 4]

8 Sep 2021

Dear Dr. Lagache,

We are pleased to inform you that your manuscript 'Tracking calcium dynamics from individual neurons in behaving animals' has been provisionally accepted for publication in PLOS Computational Biology.

Best regards,

Boris S. Gutkin

Associate Editor

PLOS Computational Biology

Lyle Graham

Deputy Editor

PLOS Computational Biology

Reviewer's Responses to Questions

**Comments to the Authors:**

Reviewer #2: The revisions have adressed my concerns. Thank you.

**Have the authors made all data and (if applicable) computational code underlying the findings in their manuscript fully available?**

Reviewer #2: None

PLOS authors have the option to publish the peer review history of their article (what does this mean?). If published, this will include your full peer review and any attached files.

Reviewer #2: No

---

## [Editor Report · Acceptance letter]

4 Oct 2021

PCOMPBIOL-D-20-01324R4 

Tracking calcium dynamics from individual neurons in behaving animals

Dear Dr Lagache,

I am pleased to inform you that your manuscript has been formally accepted for publication in PLOS Computational Biology. Your manuscript is now with our production department and you will be notified of the publication date in due course.

With kind regards,

Olena Szabo
